# Far from Health: The Bone Marrow Microenvironment in AML, A Leukemia Supportive Shelter

**DOI:** 10.3390/children8050371

**Published:** 2021-05-08

**Authors:** Stephanie Sendker, Katharina Waack, Dirk Reinhardt

**Affiliations:** Department of Pediatric Hematology and Oncology, Clinic of Pediatrics III, Essen University Hospital, 45147 Essen, Germany; stephanie.sendker@uk-essen.de (S.S.); katharina.waack@uk-essen.de (K.W.)

**Keywords:** bone marrow microenvironment (BMM), acute myeloid leukemia (AML), hematopoiesis, leukemogenesis, stromal cells, leukemic blast, therapeutic targets

## Abstract

Acute myeloid leukemia (AML) is the second most common leukemia among children. Although significant progress in AML therapy has been achieved, treatment failure is still associated with poor prognosis, emphasizing the need for novel, innovative therapeutic approaches. To address this major obstacle, extensive knowledge about leukemogenesis and the complex interplay between leukemic cells and their microenvironment is required. The tremendous role of this bone marrow microenvironment in providing a supportive and protective shelter for leukemic cells, leading to disease development, progression, and relapse, has been emphasized by recent research. It has been revealed that the interplay between leukemic cells and surrounding cellular as well as non-cellular components is critical in the process of leukemogenesis. In this review, we provide a comprehensive overview of recently gained knowledge about the importance of the microenvironment in AML whilst focusing on promising future therapeutic targets. In this context, we describe ongoing clinical trials and future challenges for the development of targeted therapies for AML.

## 1. Introduction

Acute myeloid leukemia (AML) is a malignant, hematologic disease that accounts for about one-fifth of all childhood leukemia cases. In recent decades, the prognosis of AML patients has improved thanks to advances in distinct diagnostic and therapeutic tools. Although the vast majority of children initially achieve complete remission, overall long-term survival is still limited by refractory disease and relapse, which occur in about one-third of children with AML and are linked to poor prognosis [1,2]. To date, therapy is risk-adapted but still mainly based on chemotherapeutic schemes associated with systemic toxicity. Therefore, innovative and selective therapeutic approaches with greater efficacy are urgently needed. To identify suitable, promising therapeutic targets, fundamental knowledge about leukemogenesis in AML is required.

Originating from uncontrolled proliferating immature myeloid precursor cells at different stages of maturation, the pathogenesis of AML is mainly localized in bone marrow, the major hematopoietic tissue. Physiologically, bone marrow represents a highly regenerative tissue that ensures continuous replenishment of hematopoietic cells originating from a common hematopoietic stem cell (HSC). Because the lifespan of differentiated blood cells in peripheral vessels is limited, each multipotent HSC gives rise to approximately 5 × 10^11^ cells each day, making the hematopoietic system one of the most regenerative tissues in humans [3]. Various cellular and non-cellular factors in the bone marrow microenvironment (BMM) tightly regulate this process, emphasizing the high impact of the hematopoietic microenvironment. In humans, the hematopoietic system is considered to be one of the first developing functional systems, and its localization changes throughout ontogeny. In early embryogenesis, the first hematopoietic cells are assumed to arise from the yolk sac until the second wave of hematopoiesis occurs dorsally of the aorta (referred to as the aorta–gonad–mesonephros region) (reviewed in [4]). Definitive HSCs then move to the fetal liver, which enables rapid expansion of these highly proliferative cells [5,6]. Shortly before birth, HSCs colonize the bone marrow cavity, which becomes the final hematopoietic microenvironment (BMM) [4].

Whilst hosting hematopoietic cells under healthy conditions, in leukemogenesis, this BMM is taken over by malignantly transformed and uncontrollably proliferating leukemic blasts. As these cells outcompete healthy hematopoiesis, the typical clinical presentation arises.

In analogy to the hierarchical order of healthy hematopoiesis, leukemic blasts are thought to arise from a leukemic stem cell (LSC) population that is phenotypically identified as CD34+CD38- and functionally defined by the stem cell-like properties of infinite self-renewal and leukemia engraftment in a patient-derived xenograft model [7,8]. It is noteworthy that, in immunodeficient xenotransplant models, functional LSCs have also been detected within the CD34+CD38+ fraction in AML, depending on the mouse strain, indicating that the LSC population is more heterogenous than previously suggested [9,10]. In this regard, LSCs are also referred to as leukemia-initiating or propagating cells, emphasizing their importance as the putative origin of leukemogenesis [7].

High-yielding, single-cell analyses indicate that in leukemia evolution, the acquisition of various genetic mutations leads to a complex pattern of genetically heterogenous leukemic subclones that can be either competitive or cooperative [11]. During the course of disease, environmental factors in bone marrow exert a selective pressure on leukemic cells, thereby remodeling clonal evolution [12,13]. Genetic diversity is presumed to enable the selection of resistant leukemic subclones, which in turn, could promote the evolution of relapsed and refractory disease [13].

In addition to cell-intrinsic genetic dysregulation, numerous studies have demonstrated that microenvironmental components in malignantly altered bone marrow critically impact leukemogenesis in AML (summarized in [14,15,16]). Leukemic blasts, and their neighboring microenvironmental components, seem to mutually influence each other in a bidirectional manner via complex, modified interacting pathways [16,17]. Although this is reminiscent of healthy hematopoiesis, in AML the architecture and function of the microenvironment importantly differ from those in health. In the pathological microenvironment, the altered release of mediators as well as deviating intercellular contacts lead to the formation of a malignantly modified leukemic niche that disturbs physiological hematopoiesis but fosters leukemia maintenance, thus enabling leukemia progression [17,18,19].

On the one hand, there are indications that certain changes in non-hematopoietic stromal cells may precede and, in turn, promote leukemia development [20]. On the other hand, leukemic blasts are assumed to modify the formerly healthy hematopoietic bone marrow niche to their favor, at the expense of healthy blood formation [21,22,23].

Therefore, targeting critical cellular and non-cellular interactions in the altered leukemic microenvironment could be a promising approach for achieving the major objective of establishing more effective and safer therapies that efficiently eliminate AML blasts. Redirecting this leukemia-supporting niche into a healthy hematopoietic microenvironment will require sound and comprehensive knowledge about the physiological and pathological processes in the BMM. This review aims to contribute to a better understanding of critical microenvironmental processes in healthy hematopoiesis and leukemogenesis while emphasizing promising future therapeutic directions in pediatric AML.

## 2. Hematopoietic Cells and Their Microenvironment

Physiologically, hematopoietic cells develop in a highly specialized and supportive microenvironment. As this BMM ensures lifelong continuous replenishment of the blood system, it must be tightly regulated by surrounding cellular and non-cellular elements. These microenvironmental components are constantly interconnected with each other and with hematopoietic cells, providing optimal conditions to direct HSC fate. In this context, an analogy to the “seed-and-soil” theory has been drawn: The BMM serves as fertile soil providing optimal conditions for the maturation of HSCs, which represent the seeds in the soil (summarized in [24]). As early as 1978, the first reports were published regarding the significance of specific bone marrow regions to hematopoiesis, originally referred to as the stem cell niche [25]. Based on the different structural and functional aspects of these different regions within the BMM, distinctions between different niches have been established [26]. Whereas the endosteal niche primarily consists of osteolineage cells, the perivascular niche comprises disparate sub-niches, depending on certain vascular subtypes with associated endothelial and perivascular cells (Figure 1).

Endosteal and vascular niches within the BMM consist of different hematopoietic cells (i.e., hematopoietic stem cell [HSCs] and megakaryocytes [MK]) as well as non-hematopoietic cells (e.g., osteoblasts, osteoclasts, adipocytes, sympathetic neurons with associated Schwann’s cells, distinct perivascular stromal cells [neural-glia-2 (NG2+)], C-X-C motif chemokine [CXCL]-12 abundant reticular [CAR] cells, leptin-receptor-positive [LepR+] cells, and endothelial cells) and structural components (i.e., arterioles [Art.], sinusoidal vessels [sinu.], fibronectin, collagen [Col]-1, calcium [Ca2+], and the enzymes cathepsin and matrix metalloproteinase [MMP]). Various non-cellular factors (e.g., osteopontin [OPN], thrombin cleaved OPN [trOPN], and stem cell factor [SCF)), C-X-C motif chemokine [CXCL]-12, CXCL-4, transforming growth factor beta [TGF-β], vascular endothelial growth factor [VEGF], and vascular endothelial cadherin [VE-Cad], thrombopoietin [TPO], Notch, and its ligand Jagged-1) expressed by the cellular components contribute to hematopoietic regulation in the BMM. Figure 1 was created with biorender.com (accessed on 13 February 2021).

### 2.1. Endosteal, Osteoblastic Niche Regulates Hematopoiesis

Since primitive HSCs have first been isolated from the inner bone surface, close to osteolineage cells, the endosteal region was identified as the HSC niche [27,28]. As this niche mainly consists of osteolineage cells, its importance for the regulation of HSC fate was later confirmed. Osteoblastic cells have been shown to ensure long-term maintenance of hematopoiesis by regulating quiescence in long-term (LT)-HSCs [29,30,31,32]. This form of hematopoietic cell control is mediated by various ways of direct and indirect cellular interplay.

In different mouse models, parathyroid-hormone signaling and lack of bone morphogenetic protein (BMP) receptor-1A stimulated osteoblastic cells, which in turn increased HSC numbers through Notch ligand Jagged-1- or N-cadherin-mediated cell adhesion to HSCs, respectively [31,33]. Further studies also revealed that osteoblasts essentially contribute to HSC quiescence and maintenance through N-cadherin [34,35]. However, in genetically modified mice, conditional deletion of N-cadherin revealed no significant effect on HSC function and number, suggesting that HSC quiescence may be partly influenced by but not fully dependent on this cellular interaction [36,37]. Moreover, osteoblasts promote stem cell quiescence and long-term maintenance by secretion of various soluble mediators, such as by thrombopoietin (TPO) [38,39], stem cell factor (SCF, also termed steel factor or kit-ligand) [40,41], and C-X-C motif chemokine 12 (CXCL-12) [42,43]. The latter is also considered the strongest chemotactic factor that acts on HSCs via CXCR-4, promoting recirculation of hematopoietic cells to the endosteal region [44]. Nonetheless, these factors appear to be dispensable or at least balanced by other signaling mechanisms, since deletion of CXCL-12 and SCF does not significantly compromise HSC numbers in vivo [45,46]. Moreover, osteoblasts contribute to mineralization by producing extracellular matrix (ECM) components such as collagen-1 or fibronectin [47]. The ECM protein osteopontin (OPN) has been shown to negatively regulate HSC pool size [48]. Its thrombin-cleaved form, which is found predominantly in the BMM, serves as a chemoattractant, mediating the migration and quiescence of HSCs and progenitor cells through interaction with the alpha(9)beta(1) and alpha(4)beta(1) integrins [49]. As recently been shown by Cao et al., OPN already plays a pivotal role in the fetal and newborn hematopoietic BMM (not in liver) [50]. The mentioned endosteal niche factors are cleaved by proteolytic enzymes (i.e., cathepsin-K and matrix metalloproteinase [MMP]-9) secreted by activated, bone-resorbing osteoclasts, leading to mobilization of hematopoietic progenitor cells [51]. Conversely, via osseous remodeling processes, osteoclasts cause an increase in the calcium level, which is proposed to protect quiescent HSCs through calcium-sensitive receptors [52]. In addition, bone damage has been revealed to promote HSC expansion [53], and a recent imaging study spatially allocated expanding HSCs in areas with high bone turnover [54]. Therefore, in addition to osteoblasts, osteoclasts act as noteworthy cellular components in balancing hematopoietic processes within the endosteal niche. However, distinct endosteal components and associated processes were found to be dispensable, questioning the impact of this endosteal niche in regulating hematopoiesis in the healthy BMM [37,45,55,56,57,58]. Consistently, in vivo investigations revealed that a minority of hematopoietic cells are localized close to the endosteal bone surface, but the majority are localized in the perivascular region within the BM cavity [45,59].

### 2.2. Perivascular Niches Regulate Hematopoiesis

The BMM is traversed by an intricate vascular system consisting of arterioles that adjoin the endosteum and branches, creating an intricate venous–sinusoidal network in the central bone cavity [60]. Recently, another vascular subtype was reported to connect sinusoids to arterioles in a transition zone, which has been attributed to the endosteal area rather than the central bone marrow [61]. The vast majority of hematopoietic cells have been shown to reside in the perivascular bone marrow region [43,45,59]. This niche can be sub-divided according to distinct types of vessels, which can be assigned to certain perivascular stromal cells [62] and endothelial cells (ECs) [61,63,64], forming different perivascular sub-niches critical for hematopoietic regulation in the BMM [65,66,67].

In the perisinusoidal niche, CXCL-12-abundant reticular (CAR) and leptin-receptor-positive (LepR+) cells direct the mobilization and maintenance of HSCs expressing high amounts of CXCL-12 and SCF [43,45,68]. Of note, in vivo studies revealed that HSC maintenance is necessarily regulated by SCF from LepR+ cells, whereas CXCL-12–mediated regulation by CAR and LepR+ cells appears to be dispensable [45,58].

In vivo imaging identified different subpopulations of stromal cells characterized by green fluorescent protein (GFP) expression under the control of the nestin promoter (Nes-GFP) [62]. Whereas nestin-dim stromal cells largely overlap with CAR and LepR+ cells in the perisinusoidal region, periarteriolar nestin-bright cells are reported to comprise neural-glia-2 (NG2+)-expressing cells [65]. Both contribute to the protection of HSC quiescence mainly conferred by CXCL-12 and SCF secretion [45,65].

In addition to mesenchymal-derived stromal cells, ECs serve as critical sources of regulatory niche factors and importantly contribute to hematopoietic regulation [45,58]. Given the heterogeneity of the EC population, these cells include sinusoidal-derived ECs (low expression of CD31 and Endomucin [type L] [61]) and arteriolar ECs (high expression of CD31 and Endomucin [type H]) [61,66]. Arteriolar and artery-derived ECs have been found to more abundantly express the key niche factors CXCL-12 and SCF [64]. Among ECs, arteriolar ECs are reported to almost exclusively produce SCF and thus importantly regulate HSC maintenance, whereas sinusoidal ECs expressing SCF appear dispensable [69]. When compared with sinusoidal and capillary/arteriole-derived ECs, a recently defined sca+ and vwf+ arterial EC subset has been shown to more abundantly express SCF [64], thereby representing an apparently relevant regulatory EC subset in the BMM.

Sinusoidal ECs are known to importantly contribute to the niche-dependent HSC regulation through vascular endothelial (VE)-cadherin and vascular endothelial growth factor (VEGF) receptors 2 and 3 [70,71]. This angiogenic factor (VEGF) supports the proliferation and survival of HSCs by binding to its same-named receptor (VEGFR) in an autocrine manner [72]. In vivo examination provided evidence of VEGFR2-mediated reconstitution of sinusoidal ECs, which essentially promoted regeneration and engraftment of HSCs post-transplantation [70].

Conditional deletion of VEGF-C in endothelial and stromal cells leads to decreased hematopoietic cell and niche regeneration. Conversely, exogenous administration of VEGF-C leads to improved hematopoietic niche recovery after irradiation, suggesting that VEGF is important for maintaining the HSC niche [73]. Adhesion to E-selectin, an adhesion molecule that is constitutively expressed on sinusoidal ECs, promotes HSC proliferation and activation, to the detriment of self-renewal and quiescence [74,75].

In the perisinusoidal region, a major subset of megakaryocytes is reported to positively direct HSC quiescence and maintenance by expression of mediators such as CXCL-4, TPO, and transforming growth factor beta (TGF-β)1 [76,77,78]. As part of the TGF-β superfamily, TGF-β1 functions as a pleiotropic mediator, importantly shifting the balance between HSC quiescence and proliferation in favor of the former [79,80]. To exert this regulatory function, the secreted latent form of TGF-β must first be activated by different cellular and non-cellular mechanisms in the BMM.

As the hematopoietic BMM is vastly innervated and controlled by the sympathetic nervous system [81], non-myelinating Schwann’s glial cells wrapping peripheral nerve fibers maintain the quiescent HSC pool by activating latent TGF-β [82]. However, it has been suggested that high levels of catecholaminergic transmitters stimulate HSC proliferation and mobilization [83,84]. Moreover, the circadian release of hematopoietic progenitor cells through β-adrenergic signals was shown to be promoted by neural cells of the sympathetic nervous system, induced by granulocyte colony-stimulating factor (G-CSF) [85,86].

A relevant subset of adipocytes localized in the perisinusoidal region has been recently identified as sympathetically innervated and regulated [87]. Initially, adipocytes were preliminarily believed to exert a negative influence on hematopoiesis [88]. Concordantly, in vivo studies revealed that inhibition of adipogenesis following chemotherapy-induced ablation leads to an improved regeneration of hematopoiesis [89]. However, adipocytes have been found to support hematopoietic regeneration through secretion of SCF [90] and to positively contribute to hematopoiesis through the adipokines leptin and adiponectin [91,92]. In addition, in a genetically modified mouse model, a complete lack of adipocytes led to markedly increased extramedullary hematopoiesis and alteration in CXCL-12/CXCR-4 signaling [93]. These results suggest that adipocytes are required for appropriate hematopoietic regulation in BMM. As CD169+ macrophages stimulate secretion of the retention factor CXCL-12 by periarteriolar Nes-GFP+ stromal cells, they presumably contribute to the maintenance of HSC quiescence [94].

The above-discussed results, in part, are indicative of the idea of a vascular sub-niche that plays a binary role within the hematopoietic BMM. Whereas periarteriolar localized parts seem to rather protect the quiescent HSC pool [65,95], perisinusoidal niche cells may be more likely to promote the expansion, differentiation, and mobilization of actively circulating HSCs [65,66,70]. It, therefore, seems reasonable to assume that structurally and functionally separated perivascular sub-niches broadly guarantee differential regulation of distinct hematopoietic cell subsets [95]. However, a strict distinction between these specialized sub-niches appears controversial due to anatomical and functional overlaps between these niches and has therefore not been clearly defined [59]. In addition, contrary to the long-held belief that the hematopoietic BMM consists of different niches, an increasingly accepted point of view states the existence of only one niche, comprising a periarteriolar and a perisinusoidal compartment [96]. Further research is necessary to fully elucidate the exact localization of HSCs at different stages in bone marrow niches and to clarify the specific regulatory functions of distinct niches in the direction of hematopoiesis.

### 2.3. Hypoxia in the Bone Marrow Microenvironment

Despite its pronounced vascularization, the BMM has been revealed to be hypoxic [97], with varying oxygen (O_2_) levels reported to range from 0.1% and 6% in different areas within the bone marrow [98,99]. In this hypoxic microenvironment, hematopoietic cells present a hypoxic profile, regardless of their location and cell cycle status, characterized by intracellular incorporated pimonidazole and upregulation of the transcription factor hypoxia-inducible factor (HIF)-1α [60,100,101]. Under hypoxic conditions, HIF-1α shows full transcriptional activities and has been proposed to necessarily contribute to the activation of an anaerobic glycolytic metabolic program in quiescent HSCs [100,102,103]. Hypoxic HSCs sustain their own stem cell properties through the stable expression of HIF-1α, which essentially regulates stem cell survival and quiescence [100]. In this regard, several studies also have suggested that hypoxia contributes to hematopoietic regulation in bone marrow (reviewed in [104]).

Based on indirect measurements, the endosteal niche has long been believed to be comparatively hypoxic, which was thought to support the maintenance of HSCs and subsequently influence the distribution of HSCs in the BMM [105]. However, direct in vivo measurements showed that the total extravascular oxygen tension (pO_2_) is significantly higher in endosteal regions compared to the local pO_2_ in the vicinity of nestin-negative vessels, with the lowest level found in the deeper perisinusoidal region (1.3% O_2_) [97]. More recently, O_2_ values measured in areas around HSCs and more differentiated HPCs were very similar [54]. This may suggest a lower impact of different levels of hypoxia on the microenvironmental regulation of hematopoiesis and may correspond to a dense arteriolar network, which has been found to abundantly enclose the endosteal surface [65,106].

The exact oxygen concentrations within different niches, as well as the detailed impacts on hematopoietic regulation, remain elusive and are the subject of current research studies.

## 3. Dysregulated Leukemic Bone Marrow Microenvironment

In leukemogenesis, the bone marrow is typically characterized by increased infiltration of uncontrollably proliferating leukemic cells (blasts) at the expanse of healthy hematopoiesis. It has been shown that disturbed cell-intrinsic regulation of leukemic cells is accompanied by deregulation of the bone marrow niche, suggesting a decisive role of an altered BMM in leukemogenesis (Figure 2). Of note, various studies have indicated that this malignant BMM also favors leukemogenesis by promoting immune-evading mechanisms (reviewed in Sendker et al.) [107].

The idea of a transformed microenvironment facilitating the development of hematopoietic malignancies was reported first by Dührsen and Hossfeld in 1996 [108]. Since then, several studies have investigated the malignant microenvironment. However, the exact role and contribution of the microenvironment in leukemogenesis have not yet been fully clarified. Understanding the cellular crosstalk in the leukemic microenvironment as well as its contribution to the initiation, progression, relapse, and refractoriness of AML is critical for the identification of promising therapeutic targets.

In a malignantly altered BMM, dysregulated cellular (i.e., mesenchymal stem cell [MSCs], osteoblasts, hematopoietic stem cells [HSC], sympathetic neurons, distinct perivascular stroma cells [neural-glia-2 (NG2+)], C-X-C motif chemokine [CXCL]-12 abundant reticular [CAR] cells, leptin-receptor-positive [LepR+] cells, and endothelial cells) and structural components (i.e., arterioles [Art], sinusoidal vessels [sinu], fibronectin [FN], and oxygen level [O_2_]) as well as non-cellular mediators (stem cell factor [SCF], CXCL-12, transforming growth factor beta [TGF-β], vascular endothelial growth factor [VEGF] and vascular endothelial cadherin [VE-Cad], thrombopoietin [TPO], and hypoxia-inducible factor [HIF]-1α), including related cellular interaction and signaling (vascular cell adhesion molecule [VCAM)]-1 and very late antigen [VLA)-4), constitute a supportive niche for leukemic blasts, driving pathogenesis and disease progression in acute myeloid leukemia (AML). Figure 2 was created with biorender.com (accessed on 13 February 2021).

### 3.1. Microenvironment-Mediated Malignancy Versus Malignancy-Mediated Microenvironment

In the leukemic microenvironment, dysregulations not only occur in hematopoietic cells but also in non-hematopoietic cells. This raises the question of what is first in leukemogenesis?

In non-hematopoietic stromal cells derived from patients with AML and myelodysplastic syndrome (MDS), certain non-clonal (epi-)genetic aberrations have been revealed and differ from mutations found in hematopoietic cells [20,109,110]. These results may indicate a pivotal role of mesenchymal stem cells (MSCs) in mediating leukemogenesis but do not answer the question of whether microenvironmental stromal cells facilitate disease progression rather than initiation. In mouse models, activating β-catenin mutation in osteoblasts cooperates with Foxo-1 to lead to AML development via activation of the NOTCH pathway in HSCs, which is suggestive of an initiative role of microenvironmental cells [111,112]. However, in an in vitro investigation, leukemic MSCs did not necessarily contribute to the leukemic process, although these stromal cells displayed altered phenotypic and functional features linked to the progression of leukemogenesis [113].

In contrast, multiple reports indicate that donor-derived leukemia following allogenic stem cell transplantation for different hematopoietic malignancies (reviewed by [114]) is a powerful in vivo model, suggesting that the malignant microenvironment pressures healthy donor-derived HSCs to convert into leukemic cells [114,115]. These examples favor an initiative role of the microenvironment preceding leukemic clonal transformation. Interestingly, examples of donor-derived ALL and AML were also reported post-transplantation for beta-thalassemia in a 5 year old boy [116] and for aplastic anemia in a 12 year old girl, respectively [117]. Together with the finding that a relatively small proportion (about 5%) of all post-transplantation cases of leukemia relapse originates from donor cells [118,119] and most cases relapse as acute leukemia, these observations could indicate a type of hereditary leukemic predisposition mediated by the microenvironment [114].

Contrary to the concept of a transformed microenvironment that drives leukemia according to microenvironment-mediated malignancy, an opposite but not exclusive notion assumes malignant hematopoietic cells mediate malignancy-associated changes in the microenvironment (malignancy-mediated microenvironment), as MDS cells have been shown to induce malignant/MDS-like transformation in healthy MSCs [120]. Similarly, AML-derived conditioned medium has been shown to convey leukemic alteration (regarding repressed osteogenic differentiation and proliferation) in healthy MSCs, suggesting leukemic dysregulation is instructed by leukemic cells [22].

In in vivo AML models, LSCs lead to transcriptional changes in healthy MSCs, affecting the expression of various molecular mediators involved in intercellular crosstalk [121]. Interestingly, among these distinct MSCs, a heterogenous pattern of genetic alteration has been shown and linked to a heterogeneous prognosis, presumably reflecting the clinical heterogeneity of AML [121]. A recent study involving co-culture experiments revealed that AML-derived exosomes promote interleukin (IL)-8 secretion in co-cultured stromal cells, which in turn may contribute to leukemic self-protection from chemotherapy [122]. Similarly, further studies demonstrated that AML cells convey microenvironmental disruption through the production of exosomes and vesicles (i.e., affecting stromal cell metabolism or suppressing the natural killer [NK]-cell-mediated immune response) [21,123,124].

These results suggest that leukemic cells differentially remodel the previously healthy microenvironment to their favor, indicative of a secondary induced malignant microenvironment that may further support disease progression. In this context, microenvironmental-mediated malignancy and malignancy-mediated microenvironment do not seem to mutually exclude each other, but rather to complementarily cooperate in a bidirectional manner, favoring hematopoietic insufficiency and leukemogenesis.

### 3.2. Bidirectional Interplay in a Leukemic Microenvironment

Numerous studies point to a self-reinforcing, bidirectional interplay between leukemic cells and their microenvironment that favors further disease progression (reviewed in Korn et al. 2017 [125]) and resembles a vicious cycle. In the pathological microenvironment, changes in the release of soluble mediators, such as proinflammatory cytokines, and deviating inter-cellular contacts lead to the formation of a leukemic niche at the expense of healthy hematopoiesis.

The retention of LSCs in the leukemic bone marrow niche depends on cellular interactions, which are mediated by CXCR-4/CXCL-12 signaling and other pathways. In addition, AML blasts secrete high amounts of SCF, a retention factor, that importantly attracts HSCs into less suitable areas within the leukemic BMM, from where they can only be mobilized with difficulty, which serves to impede healthy hematopoiesis [14].

Malignant, leukemic cells express certain adhesion molecules to an increased extent, such as CD44, that interact with ECs expressing E-selectin, resulting in promoted chemoresistance and upregulated viability and proliferative capacity [74]. The cellular crosstalk between vascular cell adhesion molecule (VCAM)-1 on MSCs and very late antigen (VLA)-4 (an integrin alpha4βeta1 heterodimer) on leukemic cells is reported to lead to transcriptional changes in stromal bone marrow-derived cells, causing upregulation of distinct nuclear factor kappa B (NF-κB) target genes and ultimately having a beneficial effect on leukemic cell growth, survival, and chemoresistance [126]. The interaction between VLA-4–expressing AML cells with fibronectin on MSCs has been correlated with adverse outcomes and recurrence in AML, as this interplay causes increased drug resistance mediated by intracellular activation of the phosphatidylinositol-3-kinase (PI-3K)/AKT/Bcl-2 pathway [127].

Leukemic cells increasingly produce several proinflammatory cytokines, such as Il-1 that induce surrounding ECs and stromal cells to secrete growth factors such as CXCL-12 or colony-stimulating factors, which are proposed to result in unlimited cell growth and proliferation of leukemic blasts [128]. In pediatric AML patient samples, inhibition of BMP (a molecule that belongs to the TGF-ß superfamily) using the BMP-inhibitor K02288 promoted differentiation of LSCs, while reducing viability and colony counts in vitro, suggesting the significance of the BMP-Smad pathway in pediatric AML [129].

Despite the healthy microenvironment, TGF-β restricts hematopoietic proliferation, and in the leukemic microenvironment, TGF-β has also been implicated in leukemogenesis [130]. In this regard, TGF-β has been shown to protect leukemic blasts from lethal chemotherapeutic effects and favor quiescence in LSCs, thus promoting leukemic progression [131]. Moreover, TGF-β importantly contributes to leukemia progression by dysregulating the leukemic microenvironment [132]. Whilst elevated TGF-β levels have been observed in the AML BMM [133,134], predominantly upon release by megakaryocytes (MKs) and to a lesser extent by ECs [135], other studies reported decreased TGF-β levels in AML patients’ sera [136]. However, as shown by multiple studies, TGF-β1 importantly contributes to leukemogenesis [130,132,137]. The BMM in children with Down syndrome (DS)-acute megakaryoblastic leukemia (AMKL) is characterized by marrow fibrosis [138,139], and TGF-β has been revealed as the main mediator of bone marrow fibrosis [140]. Based on pediatric patient-derived samples, Hack et al. showed that TGF-β supports fibrosis and thus significantly contributes to leukemogenesis in pediatric DS-AMKL [141].

Given that MKs and leukemic blasts abundantly express TGF-β, resulting in markedly increased TGF-β levels in the DS-AMKL BMM, and TGF-β has been shown to induce early-stage marrow fibrosis in DS-AMKL [141], TGF-β appears to play a pivotal role in the leukemogenesis of AMKL (FAB-M7) in children with and without trisomy-21. Exosomes released in AML have been revealed to contain high concentrations of TGF-β and to impact leukemogenesis and the therapeutic response [134,142]. Thereby, through secretion of exosomes, leukemic blasts remodel their environment into a leukemia-favoring niche, protecting AML blasts [21,123,143].

In AML, insulin-like growth factor (IGF)-1 and its same-named receptor IFG-R1 significantly contribute to leukemogenesis, according to evidence that IGF-1/IGF-1R signaling exerts pro-leukemic effects on AML cells in vitro via activation of downstream PI3K/Akt and the extracellular signal-regulated kinase (Erk) pathways [144,145]. In the DS-AMKL mice model, overactive IGF/IGF1R-signaling, cooperatively with the mutated lineage determining transcription factor GATA1 contribute to malignant transformation of GATA1-short mutation bearing fetal megakaryocytic progenitors, suggesting a developmental stage-specific interplay in fetal megakaryopoiesis [146]. In AML samples, increased levels of IGF-1 and associated binding proteins (IGF-BP) were shown to correlate with prognosis, which leads to the assumption that these markers have the potential to serve as predictive tools for tracing residual disease load [147]. In contrast to the stated pro-leukemic role of IGF-BP-1–6, IGF-BP-7 has a tumor-suppressive role in leukemogenesis [148].

In an MLL-AF9 AML model, it was demonstrated that leukemic infiltration leads to sympathetic neuropathy, which is presumably due to damaged β-adrenergic signaling and includes disruption of quiescent perisinusoidal nestin-positive cells, resulting in an impaired physiological niche function with reduced HSC-maintaining NG2+ periarteriolar cells but increased leukemia-supportive pericytes [149]. This process is accompanied by increased expansion of MSCs and progenitor cells, primed for osteogenic differentiation (limited to osteoprogenitor cells) [149]. In line with the latter, Battula et al. described a pre-osteoblast-rich niche that further favors the expansion of leukemic blasts [150].

In patient-derived AML cells, VEGF and its receptor VEGFR2 have been reported to be overexpressed [151] and associated with a poor prognosis in AML [152]. The angiogenic factor VEGF-C is proposed to increase the proliferation and survival of leukemic blasts while protecting them from pro-apoptotic signals [153]. Accordingly, in pediatric patient-derived AML samples, increased endogenous levels of VEGF-C were significantly linked to reduced blast elimination, reflected by increased drug resistance in vitro and higher blast count on day 15 as well as a longer time to complete remission in-vivo [154]. VEGF is promoted by hypoxia in normal and malignant cells [155]. Notably, a correlation between the hypoxia marker HIF1α and VEGF-A has been recently reported. Both are overexpressed in AML cells, but the concentration of HIF1α was significantly higher in AML-M3-derived cells than in other AML cells [156]. This could be explained by the oncogenic mutation PML-RARA in promyelocytic leukemia (M3) [157].

Hypoxia-mediated signaling through the transcription factor HIF-1α is required for the maintenance of leukemia-initiating cells in vitro [158] and in vivo [159], suggesting a pivotal role for hypoxia in sustaining leukemia. Furthermore, hypoxia promotes the recruitment of leukemic blasts expressing more CXCR-4 under hypoxic conditions through increased HIF-1α levels [160]. In addition, hypoxia not only conveys anti-apoptotic and pro-proliferative effects on leukemic blasts but also may contribute to chemoresistance [161]. In a patient-derived xenograft model (with knockdown of MIF, HIF-1α, and HIF-2α), leukemia cell proliferation was promoted by hypoxia and HIF-1α. This subsequently caused constitutive overexpression of macrophage migration inhibitory factor (MIF) in AML blasts, which in turn stimulated Il-8 expression, conferring leukemic growth and a survival advantage [162,163]. Contrary to the notion that hypoxia is a major contributor to leukemia progression, HIF-1α is also considered to have a suppressive function in leukemogenesis [164].

Though most results favor a pro-leukemic role, indicative of a highly hypoxic leukemic microenvironment, the oxygen content in AML-derived bone marrow samples is approximately as high as the pO_2_ in physiological bone marrow samples (46.05 mmHg [6.1%] and 54.9 mmHg [7.2%]) [160,165]. In contrast to most cancer cell lines, which preliminary depend on anaerobic glycolysis referred to as the Warburg effect [166]), the metabolism of some AML cells importantly depends on mitochondrial oxidative phosphorylation [167,168,169]. Furthermore, leukemic cells can switch metabolism depending on environmental conditions such as oxidative stress, which may confer a selective advantage of leukemia cells over normal hematopoietic cells [170].

### 3.3. Developmental Changes in the Bone Marrow Microenvironment

Along with the above-mentioned ontogenic changes in hematopoietic development, there are noteworthy age-related differences in the fetal, neonatal, and adult hematopoietic and leukemic BMMs. The healthy fetal liver provides a unique environment that ensures enormous HSC-expansion without malignant transformation. The results of various studies focusing on the fetal liver niche suggest an important role of fetal liver stromal cells in supporting the massive expansion of early HSCs through the expression and secretion of pivotal hematopoietic mediators, including angiopoietin-like 2 and 3, TPO, and IGF-2 (reviewed in [171,172]). The combination of the attracting factors CXCL-12 and SCF synergistically increase migration of fetal liver HSCs to its niche [173].

In the postnatal period, it has been reported that HSCs possess increased CXCL-12 expression and high cell cycling activity, which are linked to an engraftment defect that can be reversed by antagonizing the CXCL-12/CXCR-4 interplay [174]. Intriguingly, in recent studies, investigating the neonatal and adult BMM, transplantation of HSCs into neonatal mice resulted in a higher regeneration capacity than in adult BMM. Conversely, the self-renewal of LSCs, when transplanted into adult BMM, was greater than that in neonatal models, which appears consistent with prognostic differences in adult and childhood AML [175]. In addition, primitive mesenchymal stromal cells (PDGF-R+ and SCA-1+) are more abundant in the neonatal environment and secrete higher levels of pivotal microenvironmental mediators, such as Jag-1 and CXCL-12. This raises the question of whether these age-dependent microenvironmental changes could be causative for the observed changes in hematopoietic and leukemic cell engraftment, which has been confirmed by further switching transplantation experiments [175]. However, considering age as a prognostic factor, particularly in pediatric and young adult AML [176,177], the clinical prognostic impact of age-dependent changes in the leukemic BMM in children versus adults is still unknown.

## 4. Bone Marrow Directed Therapeutic Approaches in AML

Based on the high impact of this remodeled leukemia-supportive microenvironment in the pathogenesis of AML, relevant molecular structures in the BMM need to be considered as therapeutic targets for efficiently eradicating leukemic blasts and preventing relapsing/non-responding unfavorable courses of disease. To date, several cellular interactions that protect AML blasts within the highjacked leukemic microenvironment have been approved as potential therapeutic targets for disrupting this leukemia-protecting interplay.

Regarding the aforementioned physiological effects of the CXCL-12–CXCR-4 axis on CD34-positive HSC mobilization [178], administration of the first-generation CXCR-4–directed agent plerixafor (AMD3100) has been approved in combination with G-CSF to support the release of HSCs into the blood circulation [178,179]. Concurrent administration of plerixafor with chemotherapy abrogated resistance to chemotherapy, thereby decreasing the leukemia burden and increasing survival in an AML mouse model [180]. These promising anti-leukemic effects were demonstrated in patients with relapsed/refractory AML in the phase I/II study NCT00512252 [181].

The novel CXCR-4 antagonist BL8040, which has inverse agonism activity, more efficiently inhibits CXCR-4 with increased binding affinity and thus represents a next-generation CXCR-4–directed agent [182]. A combination of FLT-3 and BCL-2 antagonists has been shown to augment the anti-leukemic power of BL8040, leading to enhanced leukemia eradication and prolonged survival in in vivo trials [183].

The E-Selectin antagonist Uproleselan (GMI-1271) reduces niche-mediated chemoresistance, survival, and regeneration of leukemic cells by sensitizing them to chemotherapy while similarly reinforcing healthy hematopoiesis [184]. Based on encouraging results in a phase I/II study, a phase III study (NCT03616470) is currently assessing the clinical value of Uproleselan in relapsed/refractory AML in adults [185].

In the leukemic microenvironment, the cellular crosstalk between VLA-4 and VCAM-1 has been proposed to mediate increased blast proliferation and to reduce therapy-induced apoptosis, contributing to a poor prognosis [186]. Consistently, blockage of VLA-4 using Natalizumab, an approved VLA-4 inhibitor, leads to decreased leukemia burden and exerted promising anti-leukemic effects in vivo [187]. However, the clinical value of Natalizumab should be considered with caution according to described adverse events, especially according to the risk of leukoencephalopathy [188].

AS101 is a non-toxic VLA-4 antagonist that reconstitutes sensitivity to chemotherapy by disrupting the underlying PI3K/Akt/Bcl-2 pathway, and concomitant administration with cytarabine demonstrated promising anti-leukemic activity in pre-clinical trials [189]. The efficiency and safety of AS101 in combination with chemotherapy are currently being evaluated in a phase II clinical trial (NCT01010373) for elderly patients with MDS and non-M3 AML.

The peptide FNIII14, which disturbs VLA-4–mediated activation by disrupting the interaction between fibronectin and the beta1-subunit, exhibited synergistic anti-leukemic effects when given with cytarabine, contributing to significantly improved survival in investigated mouse models without increasing adverse myelosuppressive effects [190]. Inhibition of VLA-4 using a specific antibody (SG/17) restored susceptibility to apoptosis when given together with the chemotherapeutic agent cytarabine and rescued treated mice from residual disease [127].

Regarding the previously described leukemia-supporting impact of IGF–IGF-1R signaling in leukemogenesis in AML [145], inhibition of IGF-1R using a neutralizing antibody and the IGF-1R kinase inhibitor NVP-AEW541 abrogated this pro-leukemic effect in vitro [144]. Given that NVP-AEW541 sensitizes AML cells to etoposide-mediated toxicity in pre-clinical investigations, a first attractive therapeutic approach could be its combinatorial use together with chemotherapy [144].

Furthermore, administration of the IGF-1R inhibitor picropodophyllin (PPP) was able to abrogate the proliferative capacity of treated LSCs, which was attributed to downregulation of IGF-2 and Nanog expression [191]. Treatment with BMS-536924, a small molecule IGF-1R/insulin-receptor kinase inhibitor, reduced proliferation and induced apoptosis among AML cells through inhibition of downstream MEK1/2 and Akt signaling [192,193]. Despite these results, to our knowledge, none of these described IGF signaling-directed therapeutic tools (BMS-536924 and PPP) have yet to reach clinical phase studies. In addition, neutralizing IGF-1R antibodies and the tyrosine kinase inhibitors (TKIs) NVP-AEW541 and NVP-ADW742 have been shown to inhibit proliferation and to induce apoptosis [194,195].

Given that, in contrast to HSCs, LSCs express a markedly decreased level of the tumor-suppressing IGF-BP7, this secreted factor appears may represent a promising therapeutic agent in AML [196]. In a pre-clinical study, administration of recombinant IGF-BP7 was proven to efficiently eliminate the leukemia cell load by redirecting the leukemic phenotype towards a phenotype with increased differentiation, decreased survival, and reconstituted chemotherapy sensitivity [148].

Despite these first promising clinical results, studies assessing the utility of novel therapeutic approaches targeting the microenvironmental interaction in pediatric AML are rare, and available clinical data are limited by low incidence, emphasizing the necessity of strong international collaboration.

## 5. Conclusions

In this review, the major mechanisms of cell–cell interaction in the BMM that contribute to healthy hematopoiesis and leukemogenesis have been outlined. Microenvironmental crosstalk involves various cellular and non-cellular components that mutually influence each other in a multidirectional manner, resulting in a complex leukemia-supportive niche. Considering the high impact of this BMM in leukemogenesis, targeting such microenvironmental crosstalk may provide a promising future strategy to cure AML. The first auspicious BMM-directed therapies are already being tested, but clinical studies in pediatric AML are still limited. Further research is necessary to deepen our knowledge about the BMM and identify novel approaches to eradicate leukemic blasts.

## Figures and Tables

**Figure 1 children-08-00371-f001:**
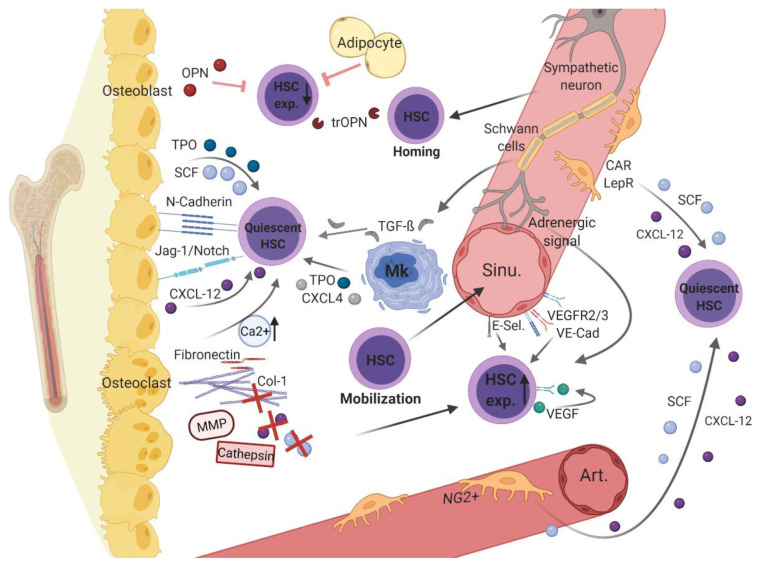
Healthy hematopoietic bone marrow microenvironment.

**Figure 2 children-08-00371-f002:**
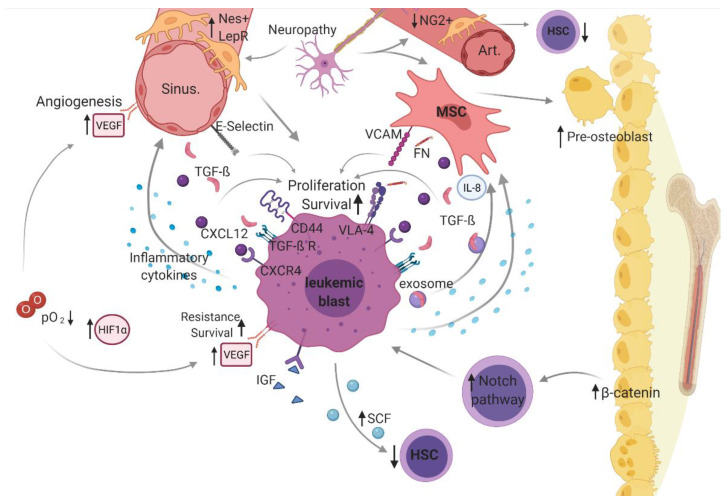
Leukemic bone marrow microenvironment.

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
