# Peer review of "Far from Health: The Bone Marrow Microenvironment in AML, A Leukemia Supportive Shelter"

_children, 2021, doi:10.3390/children8050371_

Round 1
Reviewer 1 Report
The manuscript by Sendker et al. is a well-written review article about the role of the microenvironment in pediatric AML. The manuscript has Novel insights in the pathophysiology of the bone marrow microenvironment in AML.
The bone marrow microenvironment is of importance for leukemia initiation, progression and therapy resistance. Novel therapeutic options influencing the niche are in development.
Besides, the text is clear and easy to read, the conclusions consistent with the evidence and arguments presented. And they addressed the main question posed.
Reviewer 2 Report
The authors present an interesting, comprehensive and well-written review on the bone marrow environment in AML. The discussion on the microenvironment-mediated malignancy versus malignancy-mediated microenvironment is the strongest section of the review. Given the journal, I would recommend that the authors more explicitly state whether or not data is based on experimental models that were derived from pediatric patients or adult patients. In addition, it would be very valuable to the review if the authors could provide their view (or their best guesstimate) on the differences and similarities on benign and malignant bone marrow microenvironments between adults and children. Finally, I would recommend mentioning that a relatively new and increasingly accepted standpoint is that the bone marrow niche is actually one large niche with various compartments, as opposed to different types of niches, as mentioned by the authors (see https://pubmed.ncbi.nlm.nih.gov/28363872/ for a discussion).Author Response
Please see the attachment.

Reviewer 3 Report
In this review, Sendker et al highlights the intricate intercellular communications that takes place in the leukemic niche. Overall, the manuscript content is interesting and nicely written. Few minor suggestions to correct: 1) there are several typographical errors throughout the manuscript that needs to be fixed. 2) The hypoxia section seems to belong to section 2.3.
Reviewer 4 Report
The aim of the review article is to summarise the current literature outlining the role of the bone marrow microenvironment in AML and providing evidence for its advantageous role in the survival and potential initiation of AML, including information on the most promising therapeutic targets currently under investigation.
This is a very good comprehensive review of the role of the bone marrow microenvironment in normal haematopoiesis and the changes which can occur in AML. The authors explain the microenvironment mediated malignancy and malignancy mediated microenvironment concept clearly and I agree with their conclusion of a complementary, bidirectional process. I also like that they have included a section on the developmental changes in the bone marrow microenvironment and highlights that there may be differences.
At the end of the introduction the authors state that they aim to emphasize promising future therapeutic directions in paediatric AML. All of the research data is largely from adult based in vitro or in vivo studies and the subsequent clinical trials are in adults. This is largely due to the fact that there are limited paediatric studies available and no clinical trials in children currently. I think is it important to highlight this within the manuscript and it may be worthwhile to include some specific paediatric only in vitro data examples (from the Redell group for example).
Specific comments:
Lines 55-60: The LSC population is described as CD34+CD38-. I think it is important to add information to explain that functional LSC can be found in the other CD34/CD38 quadrants in AML and that this is dependent on the immunodeficient strain of the mouse (papers for reference include Goardon et al and Tassig et al).
Line 470-471: Is this supposed to read neonatal mice as this sentence does not make sense to me?
